# Local Advantage Networks
# for Multi-Agent Reinforcement Learning in Dec-POMDPs

## Abstract

Many recent successful off-policy multi-agent reinforcement learning (MARL) algorithms for cooperative partially observable environments focus on finding factorized value functions, leading to convoluted network structures. Building on the structure of independent Q-learners, our LAN algorithm takes a radically different approach, leveraging a dueling architecture to learn decentralized best-response policies via individual advantage functions. The learning is stabilized by a centralized critic whose primary objective is to reduce the moving target problem of the individual advantages. The critic, whose network's size is independent of the number of agents, is cast aside after learning. Evaluation on the StarCraft II multi-agent challenge benchmark shows that LAN reaches state-of-the-art performance and is more scalable with respect to the number of agents, opening up a new promising direction for MARL research.

## 1 Introduction

*Reinforcement learning (RL)* (Sutton & Barto, 1998) is the branch of machine learning dedicated to learning through trial-and-evaluation by interaction between an agent and an environment. Research in RL has successfully managed to exceed human performance in many tasks including Atari games (Mnih et al., 2015) and the challenging game of Go (Silver et al., 2016).

While single-agent RL has been highly successful, many real world tasks – sensor networks (Mihaylov et al., 2010), wildlife protection (Xu et al., 2020), and space debris cleaning (Klima et al., 2018) – require multiple agents. When these agents need to act on local observations, or the problem becomes too large to centralize due to the exponential growth of the joint action space in the number of agents, an explicitly multi-agent approach is required. As such, *Multi-Agent Reinforcement Learning (MARL)* (Buşoniu et al., 2008; Hernandez-Leal et al., 2019; Shoham et al., 2007) introduces additional layers of complexity over single-agent RL.

In this paper, we focus on partially observable cooperative MARL where the agents optimize the same team reward. This setting introduces two main challenges that do not exist in single-agent RL. 1) The *moving target problem* (Tuyls & Weiss, 2012): the presence of multiple learners in an environment makes it impossible for an agent to infer the conditional probability of future states. This invalidates most single-agent approaches, as the Markovian property no longer holds. 2) The *multi-agent credit assignment problem*: to learn a policy each agent needs to determine which actions contribute to obtaining the maximum reward. While in single agent RL this problem is only temporal, as the reward can be sparse and delayed, the shared reward increases the complexity of this problem as the agents also need to determine their individual contribution.

Centralized Training with Decentralized Execution (CTDE) (Oliehoek et al., 2008a; Foerster et al., 2018; Lowe et al., 2017), has become a popular learning paradigm for MARL. The core idea behind CTDE is that even though decentralized execution is required the learning is allowed to be centralized. Specifically, during training, it is often possible to access the global state of the environment, the observations and actions of all agents allowing to break partial observability, which mitigates both the moving target problem and the credit assignment problem.

Most of the research in off-policy CTDE MARL for collaborative partially observable environments focuses on factorizing the joint Q-Value into local agent utilities such as QMIX (Rashid et al., 2018) and QPLEX (Wang et al., 2021).

In this paper, we take a radically different approach. Our *Local Advantage Networks (LAN)* algorithm learns for every agent the advantage of the best response policy to the other agents' polices. These local advantages, which are solely conditioned on the agent observation-action history, are sufficient to build a decentralized policy. In this sense, the architecture of LAN resembles independent Q-learners more than other CTDE approaches such as QMIX or QPLEX. A key element of our solution is to derive a proxy of the local Q-value that leverages CTDE to stabilize the learning of the local advantages. For each agent the Q-value proxy is composed of the sum of the local advantage with the centralized value of the joint policy. Compared to the local Q-value, LAN's proxy is able to provide better updates by breaking the partial observability and mitigate the moving target problem by integrating the changes of the other agents' policies faster. As LAN learns the local advantage function for each agent it naturally reduces the multi-agent credit assignment problem as well. LAN is also highly scalable as the centralized value network reuses the hidden states of the local advantages to represent the joint observation-action history and the number of parameters of the centralized value does not depend on the number of agents. Finally, compared to QMIX, LAN does not factorize a joint function into individual components but rather reuses a centralized network to learn the agents' advantages. This allows LAN to not have any restriction on the family of decentralized functions that it can represent as in cooperative environments the optimal policies are best response policies.

We empirically evaluate LAN against independent Q-Learners (Tan, 1993; Tampuu et al., 2015) and state-of-the-art algorithms for deep MARL, i.e., VDN (Sunehag et al., 2018), QMIX and QPLEX, on the Starcraft Multi-agent Challenge (SMAC) benchmark (Samvelyan et al., 2019). We show that on the 14 maps that compose the benchmark, LAN reaches similar performance of the SOTA in 11, surpasses the others algorithms with a large margin in 2, and under-performs in 1. In the maps with the most agents, LAN's centralized network uses up to 7 times fewer parameters than QPLEX demonstrating the scalability of our algorithm. Furthermore, in two super hard maps, LAN learns a complex strategy based on an agent sacrificing itself to lure the enemies far from its teammates, showcasing LAN's capacity to mitigate the temporally extended multi-agent credit assignment problem. This strategy allows LAN to obtain a success rate of respectively 40% and 90% on two maps where the current state-of-the-art – QPLEX – struggles to obtain any wins. LAN's average final performance on the 14 maps scores 10% more than QPLEX. We thus conclude that our approach of coordinating the learning of the local advantage functions with the centralized value-function conditioned on the agent's hidden states performs well and is highly promising, as it not only performs better, but also scales better in the number of agents in terms of the number of parameters required. LAN opens up a promising alternative research area to value factorization for learning in Dec-POMDPs.

## 2  Background

The setting considered in this paper are Dec-POMDPs (Oliehoek & Amato, 2016; Oliehoek et al., 2008a) $G = \langle \mathcal{A}, \mathcal{S}, \mathcal{U}, P, R, \mathcal{O}, O, \gamma \rangle$. At each time-step, every agent $a \in \mathcal{A}$ selects an action $u_a \in \mathcal{U}_a$ to form the joint action $\mathbf{u} \in \mathcal{U}$, where $\mathcal{U} = \prod_a \mathcal{U}_a$, that is processed by the environment to produce: a unique reward $r$ common to all agents; the next state $s' \in \mathcal{S}$; and the agents' joint observation $\boldsymbol{o} \in \mathcal{O}$, where $\mathcal{O} = \prod_a \mathcal{O}_a$, with $o_a \in \mathcal{O}_a$ the observation of agent a. As the agents cannot access the real state of the environment they condition their policy on their observation-action history $\tau_a \in \mathcal{T}_a \equiv (\mathcal{O}_a, \mathcal{U}_a)^*$, with $\boldsymbol{\tau} \in \mathcal{T}$, where $\mathcal{T} = \prod_a \mathcal{T}_a$ being the joint observation-action history. We refer to the observation-action history of an agents as its history, and the joint observation-action history as the joint history. To simplify the notations in this paper we assume that the observation function is deterministic. However the extension to stochastic observations is straightforward. With that setting, the next joint history $\boldsymbol{\tau}'$ is defined entirely by the current joint history, the joint action and the state $\langle \boldsymbol{\tau}, \boldsymbol{u}, s' \rangle$. The value, Q-value and advantage functions of the joint policy $\boldsymbol{\pi}$, which can be centralized or decentralized, are defined as:

$$V^{\boldsymbol{\pi}}(s, \boldsymbol{\tau}) = \sum_{\boldsymbol{u}} \boldsymbol{\pi}(\boldsymbol{u}|\boldsymbol{\tau}) \big[ R(s, \boldsymbol{u}) + \gamma \sum_{s'} P(s'|s, \boldsymbol{u}) V^{\boldsymbol{\pi}}(s', \boldsymbol{\tau}') \big]$$

$$Q^{\boldsymbol{\pi}}(s, \boldsymbol{\tau}, \boldsymbol{u}) = R(s, \boldsymbol{u}) + \gamma \sum_{s'} P(s'|s, \boldsymbol{u}) V^{\boldsymbol{\pi}}(s', \boldsymbol{\tau}')$$

$$A^{\pi}(s, u) = Q^{\pi}(s, u) - V^{\pi}(s)$$

We note that, if there is only a single agent a Dec-POMDP is a POMDP, and if this agent can observe the full state the POMDP is an MDP.

DQN (Mnih et al., 2013) is a popular algorithm for MDPs that learns an approximation of $Q^* = \max_\pi Q^\pi$ with a neural network parametrized by $\theta$. This $\theta$ is learned through gradient descent by minimizing $(Q(s, u \mid \theta) - y)^2$ with $y^{DQN} = r + \gamma \max_{u'} Q(s', u' \mid \theta)$. DQN uses a replay buffer to improve sample efficiency and to stabilize the learning. Dueling DQN (Wang et al., 2016) is a variant of DQN that learns both the value and the advantage, to then produce the Q-value as the sum of both instead of learning directly Q. This alternative architecture is motivated by the fact that having one part of the neural network that learns the general value of the state, and a second part that learns the effects of the actions - represented by the advantage - can be easier than learning both in the same network. DRQN uses a Recurrent Neural Network (RNN), such as a Gated Recurrent Network (GRU) (Cho et al., 2014) or an LSTM (Hochreiter & Schmidhuber, 1997), to extends DQN to partial observablity (POMDP). DQN can also be used to train independent Q-learners (Tampuu et al., 2015) for Dec-POMDPs.

## 3  Related work

Applying single agent RL algorithms to Dec-POMDPs, such as Independent Q-Learners and Independent Actor-Critic, results in poor performance due to the moving target and credit assignment problems (Tan, 1993; Tampuu et al., 2015; Foerster et al., 2018) – with the exception of stateless normal form games (Nowé et al., 2012). The replay buffer, fundamental to DQN, worsens the moving target problem as the sampled transitions are quickly outdated and off-environment as the policies evolve. As removing the replay buffer does not lead to good polices, alternatives such as importance sampling and the use of fingerprints have been explored leading to small improvements (Foerster et al., 2017). In contrast, LAN's centralized value function mitigates the moving target problem sufficiently, which enables it to take advantage of the replay buffer and to reach state-of-the-art performance.

COMA (Foerster et al., 2018) and MADDPG (Lowe et al., 2017) introduced CTDE to Deep MARL by building on single-agent actor-critic algorithms but replacing the local critic with a centralized one. In comparison, our method, LAN, is a value-based algorithm making it more sample-efficient. While LAN's joint value is also a centralized critic, it plays an intrinsically different role, as it fosters learning coordination between the local advantage functions.

In value-based methods for MARL, learning an approximate factorization of the joint Q-value into local utilities has been explored (Bargiacchi et al., 2021). In deep MARL, to ensure proper decentralized execution the factorization must follow the individual-global max (IGM) principle: the maximizing joint action of the joint Q-value must be equal to the joint action that results from maximizing the local utilities. To ensure IGM, the factorization usually enforces a monotonicity constraint, i.e., for each agent the derivative of the joint Q-value with respect to the agent's local utility is positive. VDN is the first algorithm of this kind and decomposes the joint Q-value into a simple sum. QMIX extends VDN by learning state-dependent positive weights. The state dependency broadens the family of Q-value functions that can be learned. The positive weights constraint ensures IGM. QMIX achieves good performance and improves over VDN. However, it is still limited by the monotonicity constraint. QATTEN (Yang et al., 2020) extends QMIX by using a multi-head attention (Vaswani et al., 2017) to compute the mixing weights. More recently, QPLEX extends QATTEN by transferring the IGM principle from the Q-value to the advantage function. At the cost of twice as many parameters in average and a more complex mixing network, QPLEX outperforms QMIX on SMAC. QPLEX recovers the advantage from the agents' utilities and mixes them together to obtain a centralized

advantage. In contrast, our approach learns local advantage functions that, when combined individually with the centralized value, yield the agents' Q-value proxy (one per agent). LAN does not factorize the joint advantage, leading to both better results, and better scalability with regards to the number of parameters than QPLEX

On a different direction, several algorithms focus on relaxing the monotoniticy constraint. QTRAN (Son et al., 2019) transforms the problem into an optimization problem with soft constraints. While it achieves good performance in matrix games, it fails in more complicated environments due to the loss of IGM. QTRAN uses a similar technique as LAN to represent the joint history for its centralized Q-value, however as its embeddings are not conditioned on the agent ID it uses the same mapping for different types of agents. Also, as QTRAN learns a joint Q-value its neural network has at least a linear dependency in the number of agents for the number of parameters. WQMIX (Rashid et al., 2020) extends QMIX by focusing on representing the value of good joint actions more accurately, at the expense of the accuracy for suboptimal actions. While in some maps of StarCraft WQMIX improves over QMIX, its overall performance is similar (Wang et al., 2021).

Improving multi-agent exploration or scalability regarding the action space in DecPOMDPs have been successfully explored by MAVEN (Mahajan et al., 2019) and RODE (Wang & Dong, 2020). Both works are orthogonal to ours, and while they use QMIX as a base algorithm they could also be applied to LAN. For this reason we do not include them as baselines.

Recently, MAPPO (Yu et al., 2021) and IPPO de Witt et al. (2020) proved that actor-critic-based algorithms could achieve good performance on cooperative MARL. However, they require significantly more interactions, 10 million timesteps instead of 2 million, and more computing power. Comparison with those two algorithms is also harder because MAPPO changed the state space, IPPO changed the difficulty of the enemy team, and they do not use the same version of the environment. Also, they both have different hyperparameters per map whereas the other algorithms have one set of hyperparameters for the full benchmark challenge.

## 4 Method

In this section, we present **Local Advantage Networks (LAN)** a novel value-based algorithm for collaborative partially observable MARL. LAN goes in the opposite direction of the current state-of-the-art in MARL, which focuses on factorizing the Q-value of the joint policy $Q^\pi$ into individual utilities. Instead, LAN learns for each agent the advantage of the best response policy to the other agents' policies. The local advantages are only conditioned on the own agent's history allowing for decentralized execution. The main contribution of LAN is to stabilize the learning of those advantages by leveraging CTDE to use the value of the joint policy $V^\pi$ to coordinate their learning. The centralized nature of $V^\pi$ allows to reduce the partial observability, and mitigate the moving target problem and the multi-agent credit assignment problem. By combining the local advantages with the centralized value, LAN derives a proxy of the local Q-values to simultaneously learn all components with DQN. Two key differences with a factorized Q-function are: (1) that LAN does not learn the Q-value of the joint policy, which is in fact more difficult to learn than the value ($V$), and (2) that in contrast to VDN and QMIX, LAN can represent all decentralized policies. We note that QPLEX can also represent all these policies.

We start from the observation that in a Dec-POMDP when the agents reach an optimal policy, their individual policies are best responses to the other agents' policies. Indeed, if one agent could improve its policy while the other agents polices are fixed, the joint policy cannot be optimal as the agents share the same reward. Based on this observation, LAN focuses on learning best response polices.

To better understand how to learn best response policies, we first focus on a single agent $a \in \mathcal{A}$ and assume that the joint policy of the other agents $\pi_{-a}$ is fixed. As in (Foerster et al., 2017), we derive from the Dec-POMDP $G$ a POMDP $G_a = \langle \tilde{S}, \mathcal{U}_a, P_a, \mathcal{O}_a, O_a, R_a, \gamma \rangle$, with $\tilde{S} = \langle \mathcal{S}, \mathcal{T}_{-a} \rangle$ being the original state space extended with the observation-action histories of the other agents, $P_a$ and $R_a$ are defined as follows:

$$P_a(\tilde{s}'|\tilde{s}, u_a) = \sum_{\boldsymbol{u}_{-a}} \boldsymbol{\pi}_{-a}(\boldsymbol{u}_{-a}|\boldsymbol{\tau}_{-a})P(s'|s, (u_a, \boldsymbol{u}_{-a})) \qquad R_a(\tilde{s}, u_a) = \sum_{\boldsymbol{u}_{-a}} \boldsymbol{\pi}_{-a}(\boldsymbol{u}_{-a}|\boldsymbol{\tau}_{-a})R(s, (u_a, \boldsymbol{u}_{-a}))$$

The value, Q-value and advantage of $G_a$ can then be derived as follows, with $P(\tilde{s}|\tau_a)$ the probability of being in an extended state $\tilde{s} \in \tilde{\mathcal{S}}$ when $\tau_a$ is agent $a$'s local history.

$$V^{\pi_a}(\tau_a) = \sum_{u_a} \pi_a(u_a|\tau_a) \sum_{\tilde{s}} P(\tilde{s}|\tau_a) \sum_{\boldsymbol{u}_{-a}} \boldsymbol{\pi}_{-a}(\boldsymbol{u}_{-a}|\boldsymbol{\tau}_{-a}) \big[ R(s, (u_a, \boldsymbol{u}_{-a})) + \gamma \sum_{s'} P(s'|s, (u_a, \boldsymbol{u}_{-a}))V^{\pi_a}(\tau_a') \big]$$

$$Q^{\pi_a}(\tau_a, u_a) = \sum_{\tilde{s}} P(\tilde{s}|\tau_a) \sum_{\boldsymbol{u}_{-a}} \boldsymbol{\pi}_{-a}(\boldsymbol{u}_{-a}|\boldsymbol{\tau}_{-a}) \big[ R(s, (u_a, \boldsymbol{u}_{-a})) + \gamma \sum_{s'} P(s'|s, (u_a, \boldsymbol{u}_{-a}))V^{\pi_a}(\tau_a') \big]$$

$$Q^{\pi_a}(\tau_a, u_a) = V^{\pi_a}(\tau_a) + A^{\pi_a}(\tau_a, u_a)$$

Due to the partial observability, agent $a$ needs to disambiguate the state of $G_a$ corresponding to the state of the Dec-POMDP $G$ and the joint history of the other agents. As the environment is no longer Markovian, the agent needs to base its policy on a belief over the extended state. The most straightforward way to compute this belief is to keep the full history of the agent. However, this strategy does not scale well in the number of time-steps or state space. As analyzed in the work on influence-based abstractions (Oliehoek et al., 2012), in a Dec-POMDP maintaining a belief over the subset of features that allows to locally regain the Markovian property is sufficient, using the property of d-separation. This belief is much more compact than keeping track of the entire action-observation history, and therefore offers the possibility to keep a fully sufficient representation that remains tractable. In the ideal case, the RNN's history representation will capture the belief over the d-separating features, enabling the reinforcement learning agent to learn an optimal Dec-POMDP policy. In practice of course, we aim to closely approximate such a representation, but are often uncertain of its existence, or of its size if it does exist.

Applying DQN to the single-agent POMDP $G_a$ learns, for each agent $a$, the best response policy to $\boldsymbol{\pi}_{-a}$, as the probability distribution over the relevant features $P_a$ results from executing fixed policies for the other agents. A naive solution to learn good decentralized policies would therefore be to improve each agent successively. However, this approach fails if the environment requires the agents to explore simultaneously to find the optimal policy. On the other hand, optimizing $Q^{\pi_a}$ for all the agents simultaneously, i.e., Independent Q-Learning (IQL) (Tan, 1993; Tampuu et al., 2015) also has key downsides. While IQL allows agents to explore together, it does not perform well in more complicated tasks due to the moving target problem as it ignores that the environment $G_a$ perceived by agent $a$ is shifting as $\boldsymbol{\pi}_{-a}$ evolves. So while we need agents that learn together, they need to do so in a coordinated manner.

LAN simultaneously learns best response policies and mitigates the moving target problem. These best response policies are expressed as local advantage functions that are solely conditioned on the agent's observation-action history, $A^{\pi_a}(\tau_a, u_a)$, allowing for decentralized execution. To coordinate the learning of those local advantage functions, following the CTDE paradigm, LAN leverages full information about the states and the other agents observation-action history at training time via a centralized value function $V^{\boldsymbol{\pi}}$. More specifically, LAN derives $\tilde{Q}_a^{\boldsymbol{\pi}}$ a proxy of the local Q-value $Q^{\pi_a}$ for each agent $a \in A$.

$$\tilde{Q}_a^{\boldsymbol{\pi}}(s, \boldsymbol{\tau}, u_a) = V^{\boldsymbol{\pi}}(s, \boldsymbol{\tau}) + A^{\pi_a}(\tau_a, u_a) \tag{1}$$

The proxy is constructed by summing the local advantage $A^{\pi_a}$ with the centralized value of the joint policy $V^{\boldsymbol{\pi}}$. While $\tilde{Q}_a^{\boldsymbol{\pi}}$ is not a real Q-value and it is conditioned on the full state and the joint history $\boldsymbol{\tau}$ it can be used to extract decentralized policies as the maximizing actions only depend on the agent's history $\tau_a$, as shown by equation 2. We obtain this equation by remarking that for both decomposition of $Q^{\pi_a}$ and $\tilde{Q}_a^{\boldsymbol{\pi}}$, the local and centralized values are not conditioned by the agent's actions.

$$\arg\max_{u_a} \tilde{Q}_a^{\boldsymbol{\pi}}(s, \boldsymbol{\tau}, u_a) = \arg\max_{u_a} A^{\pi_a}(\tau_a, u_a) = \arg\max_{u_a} Q^{\pi_a}(\tau_a, u_a) \tag{2}$$

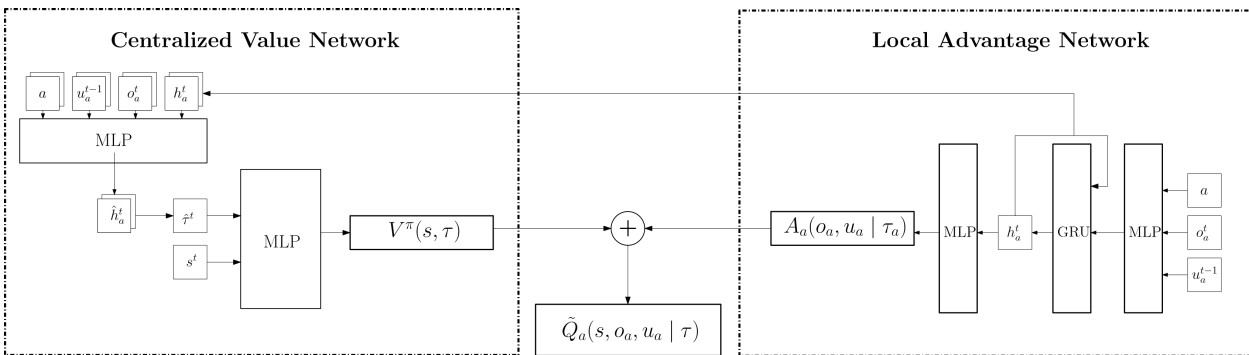

Figure 1: Architecture of LAN.

LAN uses DQN to learn $\tilde{Q}_a^{\boldsymbol{\pi}}$ for all agents $a \in A$ simultaneously. This allows LAN to learn the local advantages $A^{\pi_a}$ and the centralized value $V^{\boldsymbol{\pi}}$ in parallel by optimizing a unique loss, resulting in an efficient learning scheme. LAN's DQN target for agent $a$ is defined as follows with the subscript $t$ referring to a delayed copy of the networks to increase learning stability (van Hasselt et al., 2015).

$$y_a = r + \gamma \tilde{Q}_{t_a}^{\boldsymbol{\pi}}(s', \boldsymbol{\tau}', \arg\max_{u'_a} \tilde{Q}_a^{\boldsymbol{\pi}}(s', \boldsymbol{\tau}', u'_a)) \tag{3}$$

$$= r + \gamma [V_t^{\boldsymbol{\pi}}(s', \boldsymbol{\tau}') + A_t^{\pi_a}(\tau'_a, \arg\max_{u'_a} A^{\pi_a}(\tau'_a, u'_a))] \tag{4}$$

Compared to the local Q-value $Q^{\pi_a}$, the learning of LAN's proxy $\tilde{Q}_a^{\boldsymbol{\pi}}$ has four interesting properties that help stabilize and coordinate the learning, and give an intuition on how LAN solves the task as a whole. We note that these properties result from applying DQN to LAN's Q-value proxies to all agents in parallel, and cannot be tested independently.

First, $\tilde{Q}_a^{\boldsymbol{\pi}}$ allows to provide better update targets by breaking the partial observability. In a POMDP, the same observation-action history can be linked to different states forcing the agent to learn a Q-value that marginalizes over the possible states. In a Dec-POMDP this problem becomes harder as all the agents $a \in A$ need to marginalize over the possible states but also over the possible joint histories of the other agents $\langle s, \boldsymbol{\tau}_{-a} \rangle$ as shown by the derivation of $G_a$. By its conditioning on the next state and the joint history $\langle s', \boldsymbol{\tau}' \rangle$, LAN's DQN target does not suffer from the partial observability and can therefore provide more accurate updates. As highlighted by (Lyu et al., 2021), using a centralized target to learn a decentralized object might lead to high variance updates. The authors mention that the choice of a centralized versus decentralized critic is a bias-variance trade-off. In LAN, the value is centralized while the Advantage is decentralized. This means that the Q-value proxy used by LAN to compute the targets is not as precise as using the real Q-values. This leads to a bias, but in turn also decreases the variance.

Second, $\tilde{Q}_a^{\boldsymbol{\pi}}$ mitigates the moving target problem, which results from all the agents learning at the same time. This simultaneous learning allows the agent to explore together, which is necessary to find an optimal strategy in non-monotonic environments, but because of it the environment is constantly changing and locally loses its Markovian property. To provide meaningful updates and prevent the learning to plateau prematurely as in IQL, the updates need to reflect as closely as possible the ever changing environment. LAN achieves this thanks to the centralized value, which coordinates the learning of all the local advantages. This happens in two steps. First, as an update of $\tilde{Q}_a^{\boldsymbol{\pi}}$ results in the update of both the centralized value and the local advantage with the same transitions, a modification of a local advantage function results in a change of the centralized value. Second, as the centralized value is part of the target update of every agent's Q-value (eq. 4), the change is then propagated to the other agents' advantage.

Third, $\tilde{Q}_a^{\boldsymbol{\pi}}$ mitigates the multi-agent credit assignment problem. As the centralized value function approximates the expected return of the joint policy, the agents can easily evaluate the effect of their actions on the effective return simply by subtracting it the centralized value. This difference is then learned by the local advantages. Indeed, by applying DQN to $\tilde{Q}_a^{\boldsymbol{\pi}}$ the local advantage network of agent $a$ (eq. 5) is updated

with the following target which is similar to the one used by COMA (Foerster et al., 2018) to reduce the multi-agent credit assignment problem.

$$y_{A_a} = r + \gamma \tilde{Q}_{t_a}^{\boldsymbol{\pi}}(s', \boldsymbol{\tau}', \arg\max_{u_a'} \tilde{Q}_a^{\boldsymbol{\pi}}(s', \boldsymbol{\tau}', u_a')) - V^{\boldsymbol{\pi}}(s, \boldsymbol{\tau}) \tag{5}$$

Fourth, $\tilde{Q}_a^{\boldsymbol{\pi}}$ reduces the learning complexity of the decentralized policy. Extracting a policy from a value based algorithm is usually done by taking the maximizing action of the Q-value, or of the advantage as it has the same action ordering. Advantage and value functions have different learning complexity which are based on the environment. Indeed, compared to the advantage which learns how each action affects the return, the value learns the expected cumulative return which requires more marginalization over the different states and other agents' histories. The same reasoning in MDPs motivated Dueling DQN (Wang et al., 2016). However, the advantage cannot be learned on its own and it requires to learn the corresponding value, which suffers from both the partial observability and the moving target problem. Therefore, LAN's proxy offers a simple and efficient way to learn the local advantages without the local value.

To overcome the partial observability the local advantages networks use a GRU which learns to represent the observation-actions history into a hidden state $h_a$, with the aim to capture the necessary features to locally regain the Markov property as stated above. This hidden state is then used to compute the local advantages. LAN leverages the work done at the agent level to represent $\tau_a$ to build a representation of $\boldsymbol{\tau}$.

For each agent $a$ the centralized value network combines the id $a$ of the agent with its hidden state $h_a$, its last observation $o_a$ and its last action $u_a$ into a vector $\tilde{h}_a = [h_a, o_a, u_a, a]$. An embedding $\hat{h}_a$ of $\tilde{h}_a$ is then computed using the same feed-forward network for all agents if all the agents have similar types of observations and actions, or with a different feed-forward network per type. Finally, the centralized value network uses the sum of those embeddings $\hat{\boldsymbol{h}} = \sum_a \hat{h}_a$ to represent $\boldsymbol{\tau}$. LAN's architecture, represented in Figure 1, provides two main benefits. First, the centralized value network does not learn a second recurrent network, which are knowingly difficult to train. Second, as the embedding for all agents are computed with the same weights, the number of parameters of the centralized value network does not depend on the number of agents.

As the policies are deterministic, the local advantages should be negative with the maximizing value equal to 0. However as (Wang et al., 2016) studies, even when computing the real Q-value in single agent MDP enforcing this constraint has a negative impact on the learning. Their experiments showed that applying the following transformation to the output of the neural network provided better stability.

$$A^{\pi_a}(\tau_a, u_a) \leftarrow A^{\pi_a}(\tau_a, u_a) - \frac{1}{|U_a|} \sum_{u \in u_a} A^{\pi_a}(\tau_a, u) \tag{6}$$

In the single agent case, this results in the learned advantage to differ from the real advantage by a fixed offset. In LAN, as the centralized value is shared between all the agents, enforcing the local advantages to have a zero mean means that the offset will be shared between all the agents. As in (Wang et al., 2016), we investigated enforcing negative advantages and observed that the learning was also highly impacted by it in LAN. While sharing the offset between the agents can have a positive impact on collaboration it can also hinder the learning by adding an additional constraint on both networks. Appendix D reports LAN's performance with the mean constraint (eq. 6). Therefore, in LAN we do not apply any constraint on the output of the advantage network.

## 5 Experiments

To benchmark LAN we use the StarCraft Multi-Agent Challenge[1] (SMAC) (Samvelyan et al., 2019), a set of environments that runs in the popular video game StarCraft II. SMAC does not focus on the full game but rather on micromanagement tasks where two teams of agents - possibly heterogeneous and imbalanced - fight. A match is considered won if the other team is eliminated within the time limit. The time limits

---

[1]We use version SC2.4.6.2.69232 and not SC2.4.10. Performances are not comparable between versions.

differ per task. Each agent only observes its surroundings and receives a team reward proportional to the damage done to the other team plus bonuses for killing an enemy and winning. The action space of each agent consists of a move action to each cardinal direction, a no-op action, and an attack action for each enemy which is replaced by a heal action for each team member for the Medivacs units. The attack/heal action only affects units within range. As the agent's observation and action space are linearly dependent on the number of agents to perform well scalability is a key issue. SMAC also provides the real state of the environment, which we use as input for the centralized value. The benchmark is composed of 14 different maps that are designed to assess different aspects of cooperation. They are ranked into 3 categories: easy, hard, and super hard maps.

## 5.1 Configuration

To ensure a fair comparison, the decentralized network architecture, the version of the game, the $\varepsilon$-annealing parameters, the batch size, the replay buffer size, the use of a single environment, and the use of a unique set of parameters across all maps is consistent with the QMIX and QPLEX papers. Appendix B lists the hyper-parameters used, and Appendix D reports the results of a variation of LAN where we force the advantage to have a zero-mean as in Dueling DQN (Wang et al., 2016). In Appendix E we included a comparison of the gradient norm of the updates as a proxy for the variances updates. The training and evaluation follows the procedure described in Samvelyan et al. (2019), namely 2 million training timesteps, and evaluation of the decentralized greedy polices over 32 episodes every $10k$ timesteps. We also use the same technique as Samvelyan et al. (2019) to deal with illegal actions, such as enemies not within attack range. The advantage of those actions are set to $-\infty$ before action selection. We train LAN on at least 5 different random seeds and report the median of the battle win rate over the learning time as well as the first and third quantiles.

## 5.2 Results

We compare LAN to IQL, VDN, QMIX and QPLEX. For QPLEX we use the implementation of the authors and for the other algorithms we use the run data made available by SMAC. In the following, we present LAN's performance on 9 maps (Figure 2). The other maps are presented in Appendix C. The first row features fights between marines with an increasing number of agents and the enemy controlling more units. The second row is composed from left to right of a balanced map with 24 heterogeneous units per team, a map where 2 power-full units fight a swarm of 64 smaller enemies, and an unbalanced heterogeneous map with a medic units that as a side effect increases the action space. The last row shows the result on two super-hard maps where the baselines do not reach any wins, and a map where LAN seems to under-perform. Finally, we discuss LAN's average performance across all maps (Figure 3).

In the maps of the first row of Figure 2, two unbalanced teams with homogeneous units fight against each other, with our team composed of fewer units than the enemy: in `5m_vs_6m` 5 agents fight 6 enemies, in `10m_vs_11m` 10 agents fight 11 enemies, and in `27m_vs_30m` 27 agents fight 30 enemies. The ratio between the number of agents and the number of enemies makes the map `10m_vs_11m` easier compared to the other two. In the map `27m_vs_30m`, both the number of agents and the dimension of the observation and action space constitute a real challenge for MARL. In those three maps, LAN dominates IQL and performs on par with SOTA. First, as IQL is a natural ablation of LAN, we deduce from this experiment that the centralized value introduced by LAN does indeed help to coordinate the learning of the agents and that LAN can address the shortcomings of IQL. Second, while LAN performs on par with the SOTA, it is more scalable than QMIX and QPLEX in terms of parameters of its centralized component with respect to the number of agents (Table 1). Indeed, between `5m_vs_6m` and `27m_vs_30m` the number of agents is multiplied by 5.4 and the number of parameters of LAN's centralized value is only multiplied by a factor of 2, while for the centralized component of QMIX and QPLEX this factor is respectively 8.8 and 16.5.

The second row of Figure 2, is composed of two hard and one super-hard maps. The first one, `bane_vs_bane`, opposes two large and balanced teams of 24 heterogeneous units. We observe that while IQL easily reaches 100% of winning rate, VDN struggles to learn and QMIX fails to learn. This hints at a limitation of both monotonous mixing strategies regarding scaling to a large number of agents, supporting our claim that an alternative research direction to value factorization is needed. QPLEX is able to learn the perfect strategy

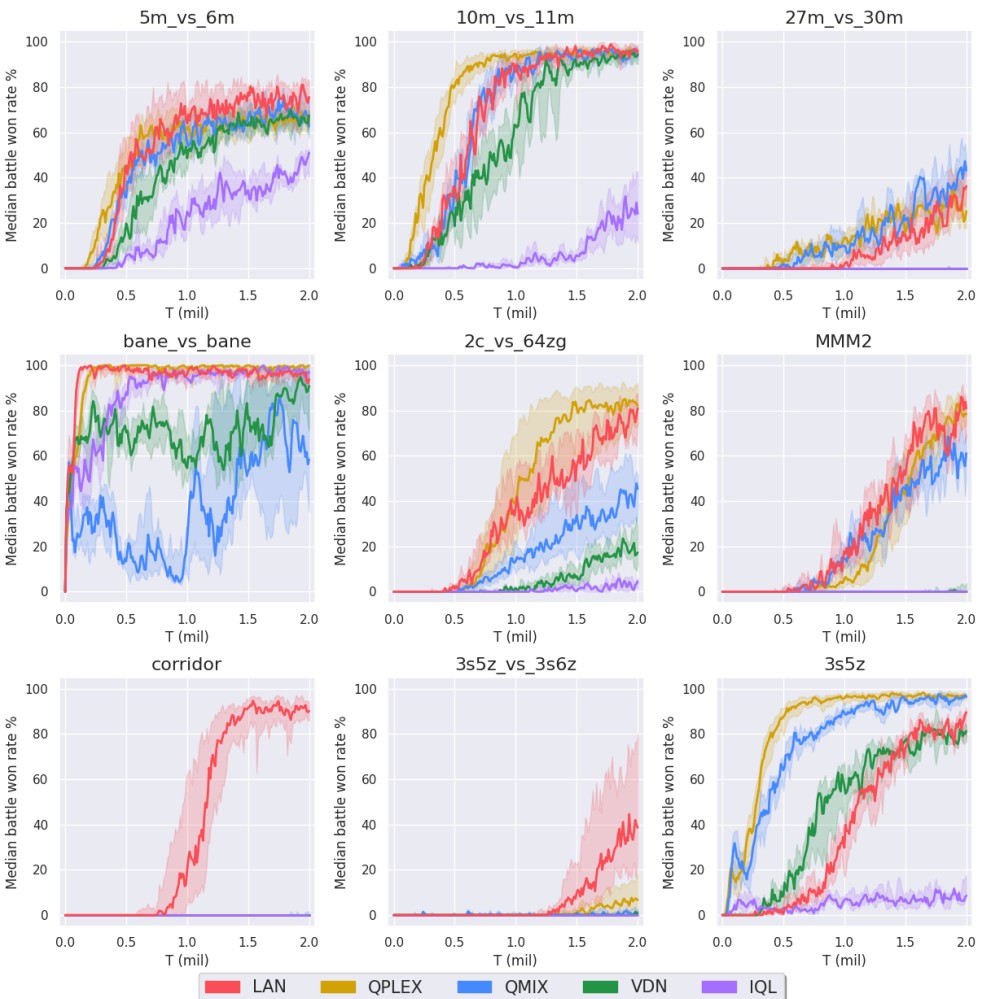

Figure 2: Median battle won rate during learning on 9 maps of SMAC. Each algorithm is run on at least 5 different seeds per map. Following the evaluation method of Samvelyan et al. (2019), we train the agents on 2 million steps and plot the median, 1st and 3rd quantiles.

at the cost of doubling the number of parameters compared to QMIX. LAN also learns to consequently eliminate the opposing team and reaches a perfect score with 5 times fewer parameters than QPLEX. The second map, `2c_vs_64zg`, matches two powerful agents against 64 weaker agents. The numerous enemies make the action space very large, with 70 actions, which is a known challenge in RL (Zahavy et al., 2018). In this map, LAN and QPLEX reach the same final performance with an 80% win rate, while QMIX and VDN score respectively around 50% and 20%. IQL struggles to learn and does not exceed a 5% win rate. The third map, `MMM2`, features two unbalanced heterogeneous teams, with the enemy team having 2 additional units, and is the only map including medical units. While IQL and VDN do not obtain any wins, QMIX and QPLEX score 60% and 80% respectively. LAN obtains the same final performance as QPLEX.

The last row of Figure 2 presents LAN's performance on 2 super-hard maps alongside the easier version of one of those maps. In the super hard map `corridor`, 6 agents of type 'zealot' fight a team 24 enemies of type 'zerlings'. While the SMAC paper claimed that the only solution for this map was to take advantage of the terrain (a spawning zone connected to a second zone by a corridor) to limit the number of enemies that can attack our agents, LAN discovered another solution. One agent lures part of the enemies to a remote location while the rest fights the remaining enemies. After killing the bait a fraction of the enemies attack our agents while the majority go through the corridor to reach the second zone. Our agents defeat

Table 1: Number of parameters (x1000) of the value function in LAN vs. the mixing network in QPLEX/QMIX for the first 4 maps of Figure 2. See Appendix A for the other maps. The dependency of the dimension of the observation and action space in the number of agents is the only cause of the difference in the number of parameters of LAN's centralized value network in the different maps

|       | 5m_vs_6m | 10m_vs_11m | 27m_vs_30m | bane_vs_bane |
|-------|----------|------------|------------|--------------|
| **LAN**   | 56 | 68  | 111 | 125 |
| **QPLEX** | 43 | 106 | 709 | 555 |
| **QMIX**  | 32 | 70  | 283 | 241 |

their attackers, and after regenerating part of their shields move to the second zone to finish off the enemies. While the current SOTA flattens to zero, LAN obtains an almost perfect score with around 90% success rate. On the next super hard map, `3s5z_vs_3s6z`, LAN learns good decentralized policies with a performance at around 40%. The only other algorithm that was able to achieve any wins is QPLEX with less than 10%. The strategy is similar as the one learned in corridor, a stalker (long-range unit) baits most of the enemy's zealots (close combat units) into targeting him. It then flees far away from his teammates and sacrifices himself so that the other agents can kill the stalkers and remaining zealots. The agents can then easily kill the remaining enemies as they are no longer protected by any long-range support. The last map of Figure 2, `3s5z`, is the balanced version of the previous map and therefore easier. In this map, LAN reaches the same performance as VDN with 80% median battle won rate whereas QMIX and QPLEX obtain 100%. This underperformance is intriguing as LAN performs better than the other algorithms in `3s5z_vs_3s6z`, the harder version of the map. By visualizing the learned policies in `3s5z` we discovered that LAN converges to two different policies: a) a basic confrontation policy which is the policy learned by QMIX and QPLEX; b) a baiting strategy identical to the one learned in `3s5z_vs_3s6z`. We also remark that LAN appears to still be learning and might converge to 100% if given more time.

The reason for LAN's performance in last two super-hard maps is its ability to train an agent to lure the enemies and to sacrifice itself for the survival of its team. We believe that this behavior is easier to discover with LAN than with the mixing algorithms because of the shared Value network, as it allows dead agents to benefit directly from the rewards scored by the other agents after their death. LAN, by focusing on learning best response policies instead of factorizing a joint Q-value, learns for each agent the policy that maximizes the team return. In the case of QMIX and QPLEX, the factorization introduces a form of individual rewards that the agents learn to maximize. If the individual rewards induced by the factorization are not aligned with the team reward, as in the baiting strategy, then the mixing strategies struggle to learn. The complex strategy learned by LAN demonstrates the capacity of LAN to mitigate effectively the multi-agent credit assignment problem.

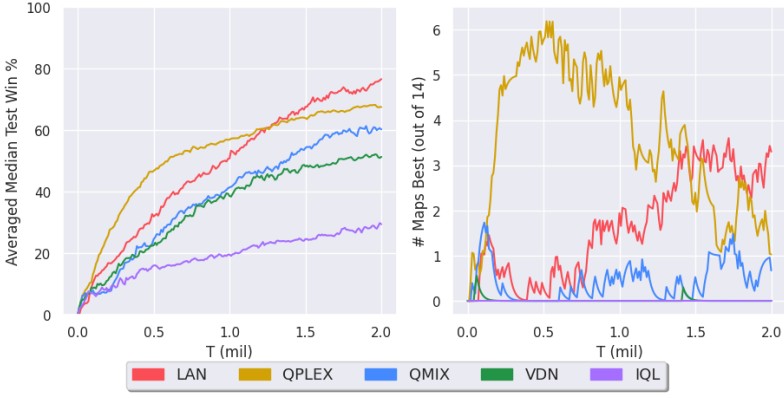

Figure 3: (Left) Averaged median test win on the 14 maps during learning. (Right) Number of maps where the algorithms are first by at least 1/32 during learning.

As in the SMAC benchmark and QPLEX papers, Figure 3 shows, on the left plot, LAN's general average performance on the 14 maps that composes the SMAC benchmark, and, on the right plot, the number of maps where each algorithm outperforms the others by a margin of at least $1/32^{\text{th}}$. IQL only achieves 30% averaged median test wins and is the best on 0 maps. This under-performance was expected as it is the only fully decentralized learning algorithm, and because it is highly vulnerable to the moving target problem. At the beginning of the learning, VDN and QMIX show similar performance, but, after $1.25e^6$ timesteps, QMIX takes the lead obtaining 60% and beating VDN by 8%. QPLEX learns faster than the other algorithms and reaches the same final performance of QMIX in just a million timesteps to obtain 67% at the end of the learning. Finally, LAN learns faster than the baselines except QPLEX, which it exceeds at around $1.25e^6$ timesteps. LAN finishes first with 77% wins. The right plot shows that LAN bests the other algorithms on 3 maps, namely `corridor`, `3s5z_vs_3s6z`, `5m_vs_6m`.

### 5.3 Credit assignment analysis

In the most difficult maps of SMAC the enemy teams have more units and the contribution of all the agents is required to win. The difference of performance between `3s5z` and `3s5z_vs_3s6z` (same team of agents but one more enemy) is a good example of that. The baiting strategy discovered in `3s5z_vs_3s6z` and `corridor` showcase the credit assignment of LAN. Indeed, while the agent that serves as bait acts at the beginning of the episode the correct behavior is reinforced even though the rewards for killing the enemies and for defeating the enemy team arrives later.

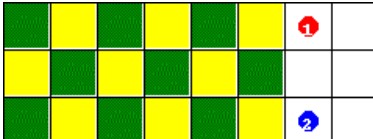
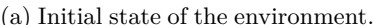
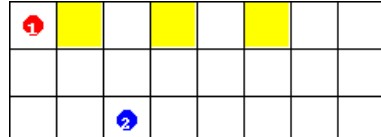

(a) Initial state of the environment.

(b) Final state of the environment reached by LAN's policy, with the first agent having eaten all the apples.

Figure 4: The Checkers environment. The green boxes are the apples, they yield +10 rewards when eaten by the first agent and +1 when eaten by the second agent. The yellow boxes are the lemons that yields −10 and −1 to the first and second agent respectively.

To further emphasize this, we performed an additional experiment on Checkers, an environments of VDN designed to asses credit assignment. In Checkers the red agent gets +10 rewards for eating apples (green) and −10 rewards for eating lemons (yellow), while the second agent gets +1 and −1 respectively. The agents receive the sum of both rewards. Each agent receives as observation its location in the map and a $3x3$ window around it. The environment finishes when there are no more apples or after 100 steps. Agent 2 needs to eat the lemons (-1 reward) that block the way for agent 1 to eat the apples (+10 reward), as shown by the initial state of the environment (Figure 4a). While both agents get the same team reward, they have distinctive roles as the second agent needs to learn that negative immediate rewards lead to a better team return. The LAN converges to the policies described above, with the 3 lemons on the top row left uneaten (Figure 4b). As this environment was designed to assess the credit assignment problem, this shows that LAN mitigates it.

In summary, LAN performs on par with the SOTA on the easy and hard maps while dominating the other methods on the super hard maps, even the ones where the other methods did not achieve any wins. LAN outperforms QPLEX by 10% in averaged performance. These results showcase LAN's performance and scalability potential, and its capacity to handle many agents and large observation and action spaces.

# 6 Conclusion

In this paper, we proposed Local Advantage Networks (LAN); a novel value-based MARL algorithm for Dec-POMDPs. LAN leverages the CTDE approach by building, for each agent, a proxy of the local Q-value composed of the local advantage and the joint value. LAN trains both networks by applying DQN to a Q-value proxy. The centralized learning allows to condition the joint value on the real state to overcome the partial observability during training. In parallel, it learns the advantages together with the joint value, to synchronize all value functions to the ever changing policies. This results in more accurate DQN targets and mitigates the moving target problem. Conditioning the local advantages solely on the agent's observation-action history, ensures decentralized execution. To ensure scalability, LAN's joint value efficiently summarizes the hidden states produced by the GRUs of the local advantages to represent the joint history. Therefore, the number of parameters of this value function is independent of the number of agents.

We evaluated LAN on the challenging SMAC benchmark where we performed significantly better or on par compared to state-of-the-art methods, while its architecture is significantly more scalable in the number of agents. In the two most complex maps, LAN was able to learn a complex strategy where one agent would sacrifice itself for the survival of the team, and therefore proving experimentally LAN's ability to mitigate the multi-agent credit assignment problem. We believe that the lean architecture of LAN for learning decentralized policies in a Dec-POMDP, inspired by influence-based abstraction, is key to learning efficiently in decentralized partially observable settings.

Most of the recent work in value-based Deep MARL for Dec-POMDP focused on improving the value factorization of QMIX. The need for a different research direction is therefore real, and LAN, by moving away from value factorization, embodies this alternative. LAN is not only able to achieve better performance than value factorization but is also more scalable parameter-wise.

# 7 Future work

In future work, we aim to explore how the history representation of the centralized value can be improved through the use of Attention (Vaswani et al., 2017) or Graph Neural Networks (Kipf & Welling, 2017). We also aim to investigate how explicit communication (Oliehoek et al., 2008b; Messias et al., 2011; Wang et al., 2020; Das et al., 2019) can be added to LAN to further improve the coordination between the agents and to improve robustness of the learned policies. We also plan to investigate how LAN's architecture might benefit MARL algorithms in settings with continuous action spaces.

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

# A   StarCraft Multi-Agent Challenge

The complete information about the SMAC benchmark can be found in the introductory paper (Samvelyan et al., 2019). Table 2 lists the 14 different maps of the challenge with the number of agents in each team and the number of parameters of the centralized part of LAN, QPLEX and QMIX. Table 3 lists the number of parameters of the centralized component of LAN, QMIX and QPLEX for the 14 maps.

Table 2: The different maps of SMAC.

| Map Name | Ally Units | Enemy Units |
|---|---|---|
| 2s3z | 2 Stalkers & 3 Zealots | 2 Stalkers & 3 Zealots |
| 3s5z | 3 Stalkers & 5 Zealots | 3 Stalkers & 5 Zealots |
| 1c3s5z | 1 Colossus, 3 Stalkers & 5 Zealots | 1 Colossus, 3 Stalkers & 5 Zealots |
| 5m_vs_6m | 5 Marines | 6 Marines |
| 10m_vs_11m | 10 Marines | 11 Marines |
| 27m_vs_30m | 27 Marines | 30 Marines |
| 3s5z_vs_3s6z | 3 Stalkers & 5 Zealots | 3 Stalkers & 6 Zealots |
| MMM2 | 1 Medivac, 2 Marauders & 7 Marines | 1 Medivac, 3 Marauders & 8 Marines |
| 2s_vs_1sc | 2 Stalkers | 1 Spine Crawler |
| 3s_vs_5z | 3 Stalkers | 5 Zealots |
| 6h_vs_8z | 6 Hydralisks | 8 Zealots |
| bane_vs_bane | 20 Zerglings & 4 Banelings | 20 Zerglings & 4 Banelings |
| 2c_vs_64zg | 2 Colossi | 64 Zerglings |
| corridor | 6 Zealots | 24 Zerglings |

Table 3: Number of parameters (x1000) of the value function in LAN vs. the mixing network in QPLEX/QMIX.

|  | LAN | QPLEX | QMIX |
|---|---|---|---|
| **2s3z** | 62 | 50 | 36 |
| **3s5z** | 74 | 90 | 60 |
| **1c3s5z** | 83 | 113 | 73 |
| **5m_vs_6m** | 56 | 43 | 32 |
| **10m_vs_11m** | 68 | 106 | 70 |
| **27m_vs_30m** | 111 | 709 | 283 |
| **3s5z_vs_3s6z** | 76 | 95 | 63 |
| **MMM2** | 86 | 136 | 85 |
| **2s_vs_1sc** | 46 | 18 | 12 |
| **3s_vs_5z** | 54 | 31 | 22 |
| **6h_vs_8z** | 61 | 59 | 42 |
| **bane_vs_bane** | 125 | 555 | 241 |
| **2c_vs_64zg** | 119 | 116 | 72 |
| **corridor** | 79 | 109 | 69 |

# B   Implementation details

We use neural networks with ReLu activation functions, to approximate the local advantage and the centralized value. To increase the learning speed and reduce the number of parameters we share the neural network weights of the local advantages between all the agents. The input of the advantage network conditions on the agent ID so that the policy can differ per agent. The advantage network is composed of a 2 hidden

layers, a 64 units feed forward network and a 64 units GRU, which is consistent with the architecture used in the SOTA algorithms to represent the decentralized utilities (Rashid et al., 2018; Wang et al., 2021).

The centralized value network (Figure 1, left) first computes an embedding of $\tilde{h}_a$ for each agent, $\hat{h}_a$, using a feed forward network of 128 units. The agents' embeddings are then merged together by summing them resulting in a joint history embedding of fixed size. This joint history embedding is then concatenated with the real state provided by the environment to create a state-history embedding. Finally, this state-history embedding goes through an feed forward network of two hidden layers of 128 units to compute the value.

We train LAN for 2 million timesteps using a replay buffer of $5k$ episodes. During training we use an $\varepsilon$-greedy exploration strategy over the local advantages, with $\varepsilon$ decaying from 1 to 0.05 over the first $50k$ timesteps. After every episode we optimize both networks twice using Adam with a learning rate of $5e^{-4}$ and without TD($\lambda$). For each update we sample a batch of 32 episodes from the replay buffer. The DQN target are computed with a target network that is updated every 200 gradient updates. We clip to 10 the norm of the gradient.

We note that LAN does not require parameter sharing, and that each type of agent could have its own model. In that case, every agent type also needs its own embedding network to compute $\tilde{h}_a$.

## C   Remaining maps of SMAC

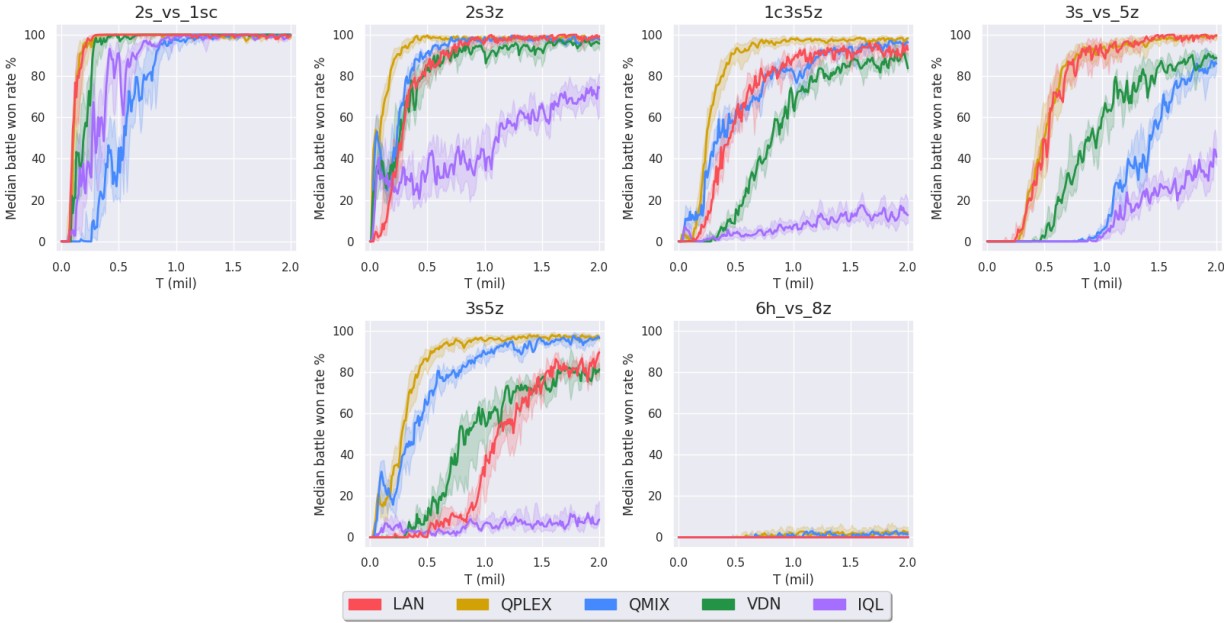

Figure 5: Median battle won rate during learning on the last 5 SMAC maps.

Figure 5 includes the 6 SMAC maps that are not included in the main paper. The first map, `2s_vs_1sc`, is an easy map and LAN learns the perfect strategy as the other algorithms do. In the second and third maps, `2s3z` and `1c3s5z`, all the algorithms but IQL learn near-optimal policies. In `3s_vs_5z`, LAN and QPLEX learn the optimal policy followed closely by QMIX and VDN that both reach around 85%. Finally, in the last map `6h_vs_8z` no algorithm is able to score any wins.

## D   Discussion regarding the advantage

Figure 6 shows the performance on all the SMAC maps with a variation of LAN called LAN mean, which applies the equation 6 . While in two maps `3s5z`, `27m_vs_30m` the mean version of LAN improves over the classical version, it degrades the performance in others other maps such as `5m_vs_6m`, `2c_vs_64zg`, and `MMM2`,

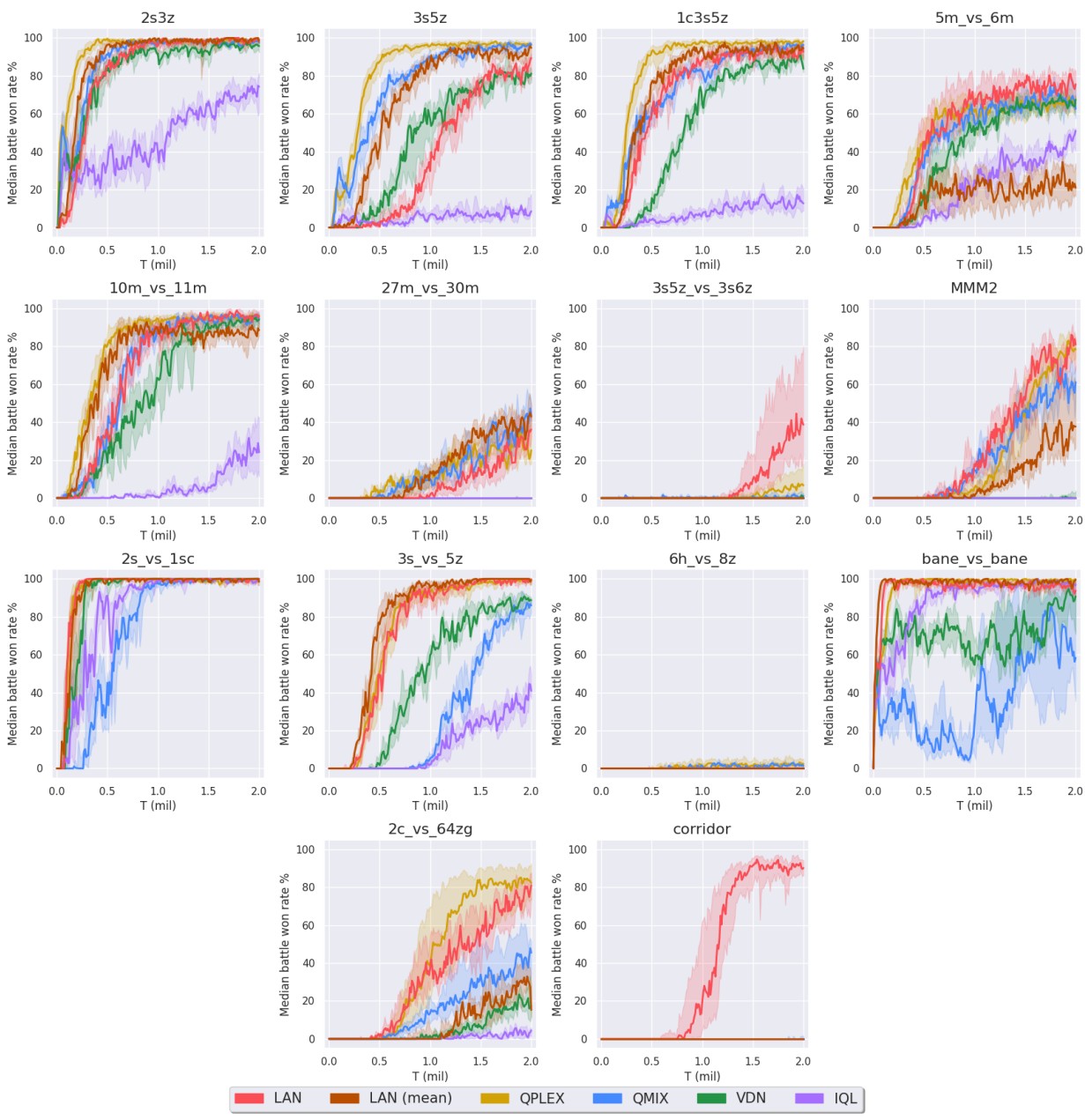

Figure 6: Median battle won rate during learning on the all the SMAC maps.

and prevents the learning in `corridor` and `3s5z_vs_3s6z`. This empirically shows that while in the single agent case the equation 6 stabilizes the learning it might not be the case when multiple agents are involved.

## E    Variance of the updates

Figure 7 shows the median gradient norm of the updates during learning on all the SMAC maps. As this information was not included in the official runs provided by SMAC, we had to run their code. We obtained similar performance as the official runs we use in Figures 2, 3 and 6. The gradient norm of LAN is generally smaller than the baselines, indicating lower variance in the updates. In some maps like `3s_vs_5z` LAN's gradient norm is bigger than some of the baselines, however the comparison also needs to consider the quality

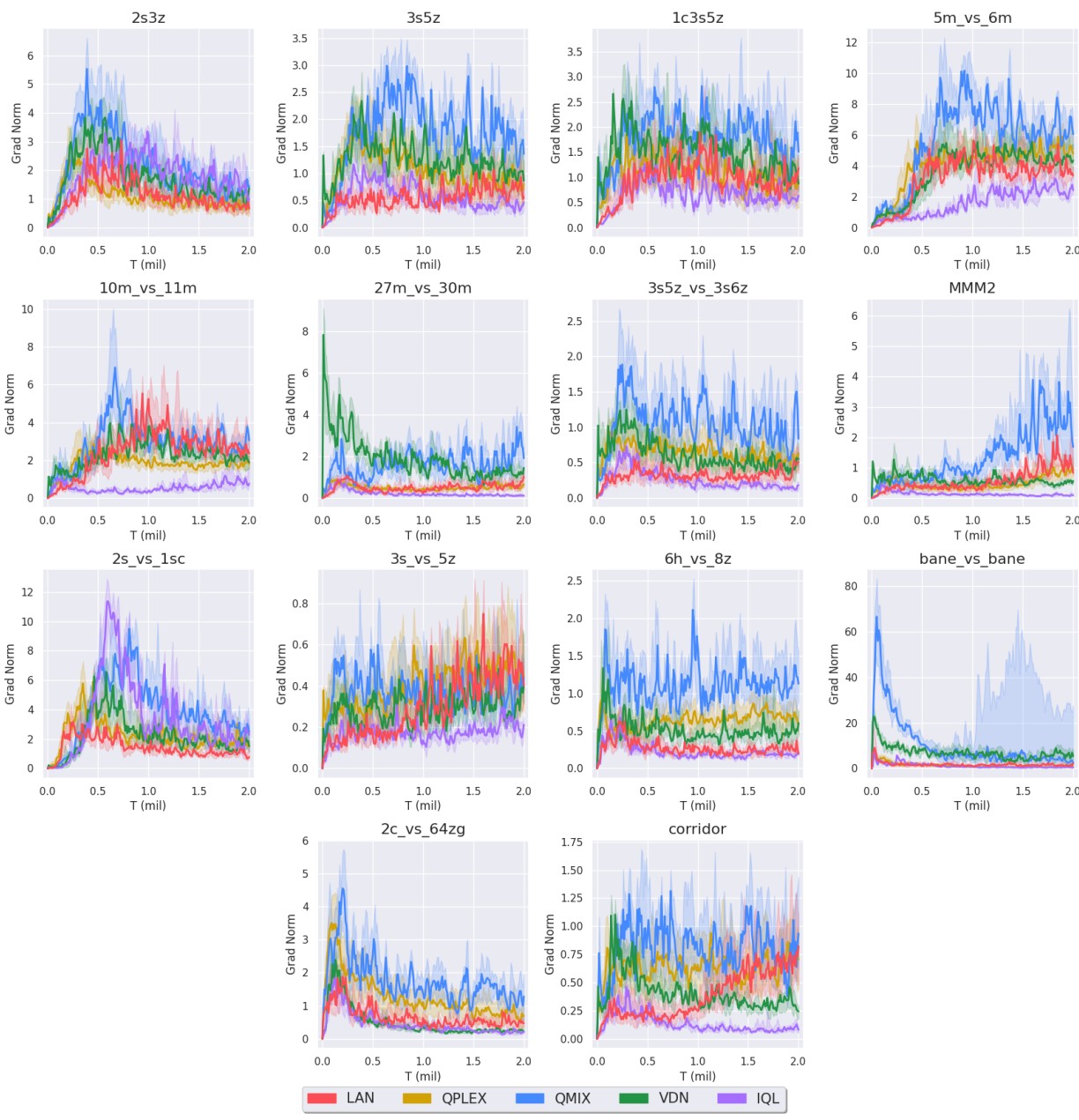

Figure 7: Median gradient norm of the updates during learning on the all the SMAC maps.

of the final policy. Indeed, in 3s_vs_5z only LAN and QPLEX reach 100% success rate and LAN has a smaller grad norm than QPLEX.

