# OpenReview forum: "Local Advantage Networks for Multi-Agent Reinforcement Learning in Dec-POMDPs"
_TMLR — Rejected by TMLR_

### Review · Reviewer_HmEE · 2022-11-09

**Summary Of Contributions:**

This paper proposes a new approach to tackle the moving target (non-stationary) problem in multi-agent reinforcement learning. The proposed method is built upon the structure of independent Q-learner, instead of widely used value factorization approaches (such as VDN and QMIX). This is a good point, as the authors provide an alternative approach for MARL problems. The basic ides is that LAN learns decentralized best-response policies with individual advantage functions for each agent, with the use a dueling network. LAN uses a centralized critic to stabilize learning. The authors conduct experiments on some of the maps from the StarCraft II Micromanagement benchmark to evaluate the effectiveness of the approach.

**Audience:**

Yes

**Claims And Evidence:**

Yes

**Requested Changes:**

I think some claims in the introduction part should be corrected to avoid overclaim. For example, it is argued that ""We show that LAN achieves SOTA performance on all the maps". However, LAN sometimes underperform state-of-the-art methods on some maps. From currently selected 8 maps in Figure 2, it is not apparent that LAN offers significant performance improvement. Could authors evaluate LAN on more maps to better analyze its advantage and disadvantage? It could be beneficial if some analysis about why LAN can sometimes underperform other baselines are described.

**Strengths And Weaknesses:**

- Strengths: The LAN method is interesting, which provides an alternative to currently well-studied value factorization approach. LAN tackles the problem from another perspective and is built upon independent Q-learners. The paper is clearly-written and easy to follow.

- Weaknesses: My main concern for the paper is its experimental results. In Figure 2, 8 maps of SMAC are studied to evaluate the effectiveness of LAN. However, on 6 out of 8 maps, LAN does not outperform state-of-the-art methods. Therefore, it is not well-supported to claim that "We show that LAN achieves SOTA performance on all the maps" in the introduction part. It will be much more convincing to also include standard deviation in Figure 3.

    - Minor: In addition, I think it could also be beneficial to discuss the connection with and novelty over QPLEX in Section 3 which also utilizes the dueling network and advantage function.

---

> ### Author Response · Authors · 2022-11-18
> **Response to Reviewer HmEE**
>
> We thank the reviewer for his time and suggestions.
>
> The benchmark is composed of 14 maps and due to space limitation we only included 8 of them in the main paper. The 6 remaining maps are discussed in Appendix C. Appendix D also includes a comparison of LAN and LAN mean, an alternative version of the algorithm, on  the 14 maps. We will add the remaining plots to the main paper. Figure 3 was introduced by the paper that presented the benchmark [1] and was also included in QPLEX [2] in its current form. As requested we will add the averaged 1st and 3rd quantiles to the left plot of Figure 3.
>
> We agree with the reviewer that our claim could be misinterpreted as we do not surpass the other algorithms in every map but rather have similar final performance to the best algorithm in each map and surpass, with a big margin, the other algorithms in two super difficult maps (corridor and 3s5z_vs_3s6z). The only map where LAN performs worse than the best is 3s5z because the algorithm is still learning after 2 million steps and depending on the runs LAN finds two different strategies. We refer the reviewer to appendix C for a more detailed explanation of LAN’s performance in that map. Our claim was motivated by the fact that the 14 maps form a unique benchmark with a global scoring (averaged median win rate), and that LAN suprasses the previous SOTA by 10% on that metric. We will make sure to rephrase this carefully in the introduction to clarify this point.
>
> Regarding the use of a dueling structure in both LAN and QPLEX [2], the two algorithms leverage the separation of Q-Values into Value and Advantages for different reasons. On one hand, LAN learns a unique centralized Value that when summed with an agent’s local Advantage returns the agent’s Q-Value proxy. This leads to a Q-Value proxy for each agent. On the other hand, QPLEX computes for each agent a local Q-Value that is first transformed with state dependent weights and separated into agentwise Values and Advantages. They are then mixed  between the agents  separately leading to a global Value and Advantage that are then summed to obtain a centralized Q-value.
>
> [1] Mikayel Samvelyan, Tabish Rashid, Christian Schroeder de Witt, Gregory Farquhar, Nantas Nardelli, Tim G. J. Rudner, Chia-Man Hung, Philip H. S. Torr, Jakob Foerster, and Shimon Whiteson. The StarCraft
> Multi-Agent Challenge. Proceedings of the International Joint Conference on Autonomous Agents and
> Multiagent Systems, AAMAS, 4:2186–2188, 2 2019.
> [2] Jianhao Wang, Zhizhou Ren, Terry Liu, Yang Yu, and Chongjie Zhang. QPLEX: Duplex Dueling Multi-Agent Q-Learning. International Conference on Learning Representations, 8 2021

---

### Review · Reviewer_3z8K · 2022-11-11

**Summary Of Contributions:**

The authors propose a value-based method for tackling the Dec-POMDP multi-agent setting by advocating for further decentralizing of the value function, unlike the standard centralized training, decentralized execution (CDTE) setting. Specifically, they propose to learn local advantages that take into account the other agents’ policies. Unlike centralized methods, they do not learn the Q-value of the joint policy and instead learn the state-only value function, and claim that unlike VDN and QMIX it can represent all decentralized policies.


**Audience:**

Yes

**Claims And Evidence:**

No

**Requested Changes:**

Related work should come before the experiments, or a more thorough explanation of the differences between algorithms used as baselines. Going into the experiments, I have no idea what QMIX, QPLEX, and VDN are. Are they also based on IQL? Otherwise, how is this a fair comparison?
Analysis should be done on the predicted Q/V values of the various methods to confirm that other algorithms are actually extrapolating more poorly because they train a centralized Q and is the reason for the improved performance by LAN.
What about an ablation where you learn a centralized Q and V to compute the advantages?

More description is needed in Figure 2.

Nits:
Word -> world: 2nd paragraph of intro.
This result in -> this results in (under Eq. 6).
Wining -> winning in section 4.2 2nd paragraph.


**Strengths And Weaknesses:**

Strengths:
The proposed method makes sense intuitively and the authors lay out clear claims of what LAN is capable of:
Claim: By proposing to use a centralized V instead of Q, the authors suggest that this becomes lower variance
Claim: Mitigates the moving target problem by propagating a change in any agent’s policy to the advantage functions over all agents.
Claim: Mitigates multi-agent credit assignment problem because can evaluate effect of actions by subtracting the centralized value.
Claim: Reduces learning complexity by learning local advantages without the local value

Weaknesses:
There are no analysis experiments backing up each of the claims. One suggestion is to measure the variance of the updates to check that using centralized V is actually lower variance.
It’s not clear the results are significant - LAN outperforms baselines in corridor and 3s5z_vs_3s6z but not in other environments - marking which results are statistically significant would be helpful.

---

> ### Author Response · Authors · 2022-11-18
> **Response to Reviewer 3z8K**
>
> We thank the reviewer for his time and suggestions. As multiple reviewers suggested that we perform ablations we answered this question in a different thread, we therefore invite you to read our answer there. We also followed your suggestion to look into the variance of the updates represented by the gradient norm of the update. The results corroborate our claim and are included in the other thread.
>
> All the experiments are run on at least 5 different random seeds, and our evaluation procedure is the same as the papers introducing the benchmark and of QPLEX. This also includes Figures 3 which appears in the other papers.
>
> We understand the point of the reviewer that there should be more explanation about the baselines before the experiment. We are in the process of modifying the paper to address this issue, alongside  additional description in Figure 2, and correction of the typos.

---

### Review · Reviewer_YcBP · 2022-11-14

**Summary Of Contributions:**

The authors are introducing a simple approach to the MARL problem, learn a centralized value function shared by all agents and individual advantage functions. They perform experiments on the standard SCII micromanagement benchmark tasks and show, in particular, better results on the "super-hard" maps.



**Audience:**

Yes

**Broader Impact Concerns:**

No particular concerns

**Claims And Evidence:**

Yes

**Requested Changes:**

Please introduce some simple illustrative tasks where free-riding is the central issue and analyze how the new approach is solving it and performs credit assignment.

**Strengths And Weaknesses:**

I would have expected this to be useful for sample complexity, which can help on the hardest SCII micromanagement maps where the authors do show their most impressive results. However, the authors also say that the approach deals with the credit assignment problem that is central to the MARL problem. I am not sure how well the SCII tasks tests for the ability to deal with issues like free-riding that motivated the VDN approach that many of the other approaches are based upon, and that approach was introduced with some tasks testing for free-riding and performed analysis on how the credit assignment work. It would have been good if the authors has provided some experiments that really test and analyse how the new approach is performing credit assignment (like displaying the individual advantages for an episode in some illustrative situation).

The paper is overall well presented and introduces a simple approach that could be promising, into an area that has started to generate huge complexity.

---

> ### Author Response · Authors · 2022-11-18
> **Response to Reviewer YcBP**
>
> We thank the reviewer for his time and suggestions. As multiple reviewers suggested that we perform ablations on the claims that we make in the paper, among which the claim about mitigating the credit assignment problem, we decided to answer this question in a separate thread. As you suggested we applied LAN to an environment from VDN that was designed to test credit assignment. We added in the supplementary material an animated image of the learned policies in this environment.

---

### Review · Reviewer_uvcs · 2022-11-14

**Summary Of Contributions:**

Instead of learning a factorized value function, this paper extends the idea of dueling DQN to multiagent settings and learns a local advantage function for each agent. Besides, a global state value function is learned to stabilize training and mitigate the moving target problem. Experiments show the proposed algorithm achieves competitive performance compared with previous methods and some promising results considering the scalability.

**Audience:**

Yes

**Broader Impact Concerns:**

Not applicable.

**Claims And Evidence:**

Yes

**Requested Changes:**

I suggest the authors revise the manuscript considering

1) More technical details and theoretical analysis about Equation (2).

2) Ablation studies to support the proposed claim.

3) One thing I found in the previous paper [1] is that legal action mask is critical in SMAC, so I wonder how action masks are used in this paper (when selecting an action and updating the value network).

4) I recommend authors do experiments on the latest version of SMAC since it has been released for more than 2 years and most recent works are conducting experiments on the new version.

[1] Action semantics network: Considering the effects of actions in multiagent systems. ICLR 2020.

**Strengths And Weaknesses:**

Strengths:

1) Learning a global state value function is more scalable regarding the number of agents.

2) Experiments also show promising results regarding the scalability of the proposed method.

Weaknesses:

1) More technical details and theoretical analysis should be provided about Equation (2). There are some assumptions behind Equation (2) that make this equation establish. Also, add more discussion about the claim 'LAN does not have any restriction on the family of decentralized functions'.

2) Ablations should be added to the experiment section, explaining the functionality and supporting the claim of four properties.

---

> ### Author Response · Authors · 2022-11-18
> **Response to Reviewer uvcs**
>
> We thank the reviewer for his time and suggestions. As multiple reviewers suggested that we perform ablations we answered this question in a different thread, we therefore invite you to read our answer there.
>
> Equation (2) links the maximizing action of the Q-Value proxy with the local Advantage and the local Q-Value of the same agent. The first equality, between the Q-Value proxy and the local Advantage, results from the definition of the proxy as the sum of the local Advantage and of the centralized Value (eq. 1) and the fact that the centralized Value is not conditioned on the action. The second equality, between the local Advantage and the local Q-Value, results from the same observation on the local Q-Value.
>
> The claim “This allows LAN to not have any restriction on the family of decentralized functions that it can represent as in cooperative environments the optimal policies are best response policies.” is supported by two elements. First, LAN does not enforce an architectural constraint to the network as opposed to QMix. Second, all Q-Value can be separated into a Value and an Advantage (by definition).
>
> To avoid the selection of illegal actions we use a standard approach which consists of masking the illegal actions before performing the max/argmax. The different baselines also use this technique (see the code of the paper introducing the benchmark [1]). The paper mentioned by the reviewer develops interesting ideas to deal with large action spaces. Its contributions are orthogonal to LAN and while showcased on QMIX they could be extended to LAN. However, with LAN we are interested in proposing an alternative to value factorization for value based MARL and therefore only compare to base architectures.
>
> As the new version of SMAC changes significantly the difficulty of some maps we would need to perform another hyperparameter tuning for LAN, but more importantly also for all the baselines we use. This would require a significant amount of time and compute power as SCII is slow compared to other environments, and we believe would not provide any significant scientific improvement.
>
> [1] Mikayel Samvelyan, Tabish Rashid, Christian Schroeder de Witt, Gregory Farquhar, Nantas Nardelli, Tim G. J. Rudner, Chia-Man Hung, Philip H. S. Torr, Jakob Foerster, and Shimon Whiteson. The StarCraft
> Multi-Agent Challenge. Proceedings of the International Joint Conference on Autonomous Agents and
> Multiagent Systems, AAMAS, 4:2186–2188, 2 2019.
> Code: https://github.com/oxwhirl/pymarl

---

### Author Response · Authors · 2022-11-18
**General Response**

We would like to thank all the reviewers for their time and their insightful comments that will help improve the quality of the paper.

Multiple reviewers requested additional information on the claims we make in the paper. We answer this question here. The other points are addressed individually.

In the paper we claim that LAN’s proxy provides mitigation on partial observability, credit assignment problem, the moving target problem, and also reduces the learning complexity. While we would love to perform ablations, those elements are not individual add-ons but rather consequences of applying DQN to the Q proxy for all agents in parallel. For instance reviewer 3z8K proposed an ablation where we would learn a centralized Q and V to compute the advantages. While learning a centralized V is possible, the curse of dimensionality makes the learning of a centralized Q intractable as the number of agents grows without resorting to techniques like factorization. Even then by learning a centralized Q, we would obtain a centralized Advantage that we would not be able to use for decentralized execution. As an alternative, we provide additional explanations to support each claim. We will also add them in the paper.

For the **moving target problem**, we would like to highlight that IQL is a natural ablation of LAN: by trading the centralized value of LAN for individual local ones we obtain IQL. As shown in the experiments LAN surpasses IQL in every map. Combined with the fact that IQL is known to suffer from the moving target problem [1], this indicates that the centralized value does indeed help mitigate that problem.

For the **credit assignment problem** we provide the following analysis based on the experiments SCII. In the most difficult maps of SMAC the enemy teams have more units and the contribution of all the agents is required to win. The difference of performance between 3s5z and 3s5z_vs_3s6z (same team of agents but one more enemy) is a good example of that. The baiting strategy discovered in 3s5z_vs_3s6z and corridor showcase the credit assignment of LAN. Indeed, while the agent that serves as bait acts at the beginning of the episode the correct behavior is reinforced even though the rewards for killing the enemies and for defeating the enemy team arrives after.

To further emphasize this, we performed an additional experiment on one of the environments of VDN (checkers) as requested by reviewer YcBP. In Checkers the red agent gets +10 rewards for eating apples (green) and -10 rewards for eating lemons (yellow), while the second agent gets +1 and -1 respectively. The agents receive the sum of both rewards. Agent 2 needs to eat the lemons (-1 reward) that block the way for agent 1 to eat the apples (10 reward). While both agents get the same team reward, they have distinctive roles as agent 2 needs to learn that negative immediate rewards lead to a better team return. LAN converges to the policies described above, with the 3 lemons on the top row left uneaten. As this environment was designed to assess the credit assignment problem, this shows that LAN mitigates it. We added in the supplementary material an animated image of the learned policies.

About the **learning complexity** we explain that LAN is less complex because it only learns a local advantage by agent and not the corresponding local value. Advantage and value functions have different learning complexity which are based on the environment. Indeed, compared to the advantage which learns how each action affects the return, the value learns the expected cumulative return which requires more marginalization over the different states and other agents' histories. The same reasoning in MDPs motivated Dueling DQN [2].

For mitigating the **partial observability**, we explain that the centralized value helps provide better target updates while not being as precise as Q which can be beneficial from the point of view of the update variance. In [3] they analyze how using a centralized network as a target for the learning of a decentralized network usually increases the variance of the update. The authors mention that the choice of a centralized vs decentralized critic, the Value in LAN, is a bias-variance tradeoff. In LAN, computing the targets with the Q-Value proxy leads to a bias compared to using the real Q-Value, which in turns decreases the variance. As reviewer 3z8K suggested we looked into the variance of the updates represented by the gradient norm of the updates. We added the plot in the supplementary materials. The gradient norm of LAN is generally smaller than the baselines, indicating lower variance in the updates. In some maps like 3s_vs_5z LAN’s gradient norm is bigger than some of the baselines, however the comparison also needs to take into account the quality of the final policy. Indeed, in 3s_vs_5z only LAN and QPLEX reach 100% success rate and LAN has a smaller grad norm than QPLEX.

We hope to have answered the reviewers' concerns.

---

> ### Author Response · Authors · 2022-11-18
> **References of the general response**
>
> [1] Guillaume J. Laurent, Laëtitia Matignon, Nadine Le Fort-Piat. The world of Independent learners is not Markovian.. International Journal of Knowledge-Based and Intelligent Engineering Systems, IOS Press, 2011, 15 (1), pp.55-64. ff10.3233/KES-2010-0206ff. ffhal-00601941f
>
> [2] Ziyu Wang, Tom Schaul, Matteo Hessel, Hado Van Hasselt, Marc Lanctot, and Nando De Freitas. Dueling Network Architectures for Deep Reinforcement Learning. In 33rd International Conference on Machine Learning, ICML 2016, volume 4, pp. 2939–2947, 11 2016. ISBN 9781510829008
>
> [3] Xueguang Lyu, Yuchen Xiao, Brett Daley, and Christopher Amato. Contrasting Centralized and Decentralized Critics in Multi-Agent Reinforcement Learning. Proc. of the 20th International Conference on Autonomous Agents and multi-agent Systems (AAMAS 2021), 2 2021.
>
> [4] Jianhao Wang, Zhizhou Ren, Terry Liu, Yang Yu, and Chongjie Zhang. QPLEX: Duplex Dueling Multi-Agent Q-Learning. International Conference on Learning Representations, 8 2021
>
> [5] Peter Sunehag, Guy Lever, Audrunas Gruslys, Wojciech Marian Czarnecki, Vinicius Zambaldi, Max Jaderberg, Marc Lanctot, Nicolas Sonnerat, Joel Z. Leibo, Karl Tuyls, and Thore Graepel. Value-decomposition networks for cooperative multi-agent learning based on team reward. In Proceedings of the International Joint Conference on Autonomous Agents and Multiagent Systems, AAMAS,

---

### Author Response · Authors · 2022-12-02
**Paper update**

We updated the paper to answer the reviewers' feedback.

---

### Decision · Action_Editors · 2023-01-04

**Recommendation:** Reject

**Comment:**

**Summary:** This paper considers learning in decentralized POMDPs with cooperative agents. If each agent learns separately in a completely decentralized manner, as in Independent Q-Learning, the performance would be poor due to the challenging moving target problem caused by the change in other agents' policies and the multi-agent credit assignment problem. There are approaches based on Central Training Decentralized Execution (CTDE) that address this issue, but they are based on specific factorization of the action-value function.

This paper proposes Local Advantage Networks (LAN), a new approach that is not based on the factorization of the action-value function. It maintains a central state-value function, which has access to the true state of the underlying MDP, and uses it along local advantage functions, which are trained in a decentralized fashion for each agent, to compute a proxy action-value function. This proxy serves as the target for DQN-like algorithm. The paper shows empirical evidence showing that the suggested method is often on par with other state of the art methods, and in certain domains, much better than them.


The paper mentions four reasons why this formulation is helpful: better update targets (BT), mitigating the moving target problem (MT), mitigating multi-agent credit assignment (MACA), and reducing learning complexity (LC). I will refer abbreviations later.


**Evaluation:** The reviewers appreciate several positive aspects of this paper. They agree that the method is interesting and provides an alternative to current factorization approaches. They also believe that the experiments show promising results, especially on "super hard" maps, where the difference with other methods is significant.


On the other hand, the reviewers have raised several concerns, including the insufficiency of evidence of some claims, and the statistical significance of results. I shall explain these in more detail. When I read the paper, I also found some issues with its clarity and preciseness.


The reviewers are divided in their final decision. We have two Leaning Accepts (3z8K and YcBP) and two Leaning Rejects (HmEE and uvcs). None of the positive reviewers are strongly advocating the work.

I read the revised paper and went through the reviews to see whether the concerns have been addressed adequately or not. I will write detailed comments below. As a high-level summary, I should say that some of the major concerns have been addressed. One remaining major issue is related to the evidence for why the suggested method works (BT, MT, MACA, and LC claims above). The authors have partially addressed them, but their evidence is not uniformly strong and satisfying.

Overall, I believe the idea presented in this paper is promising and the authors have considered several aspects of it. But given that the reviews are mixed and several concerns are beyond the level of minor edits, I believe the paper is not ready to be accepted as is and requires another round of careful reviewing. I encourage the authors to revise the paper and resubmit.



**Detailed Description of Concerns:**
I explain what the main concerns are and whether they have been addressed or not.


- The paper claims four properties/aspects of the method (BT, MT, MCA, and LC), but does not provide ablation studies or analysis backing up each claim. [Reviewers 3z8K and uvcs]

My evaluation: The authors have partially addressed this concern in their response. For example, they added Appendix E that shows the gradient norm of the updates, partially answering the BT claim (but is that the best way to provide evidence for that claim?). They added a new environment to support the MACA claim.

Their answer to LC is less satisfactory. It is not clear whether learning complexity refers to "sample complexity" or something else. The paper has statements such as "Advantage and value functions have different learning complexity which are based on the environment" (just after Eq. 5). While this might be true, not much empirical or theoretical evidence is provided. Moreover, it is not clear how LAN benefits from it.

I encourage the authors to focus on this aspect of their paper because even the positive reviewers were not completely satisfied by this.


- Statistical significance of results is unclear. [Reviewer 3z8K]

My evaluation: LAN clearly outperforms baseline in two domains (corridor and 3s5z_vs_3s6z), but the difference in other domains is not clear, as the variances of all methods are high. The claim such as LAN's average final performance on the 14 maps scores 10% more than QPLEX (the last paragraph of Section 1) is not meaningful without proper statistical hypothesis test. In other words, is 10% statistically significant without considering the variance?

Also the paper mentions that the experiments are run with at least 5 different random seeds. Why not providing the exact number? Is it 5 or 50 runs?

The paper shows 1st and 3rd quartile (they refer to it as quantile, but I assume they mean 4-quantile or quartile). If it is 5, 1st and 3rd quartile are the second smallest and second largest values. Can one reliably estimate them with only 5 data points?


- Figure 3 does not have standard deviation information. [Reviewer HmEE]

My evaluation: The authors have promised to fix this, but they have not yet.





- More technical discussion around Eq. (2). [Reviewer uvcs]

My evaluation: This is addressed in the rebuttal, and the paper is also revised to reflect it.


- The paper claimed that "LAN does not have any restriction on the family of decentralized functions". [Reviewer uvcs]

My evaluation: The authors clarified this in their rebuttal, but I agree with the reviewer that its meaning is unclear when the claim was made in Introduction. Please revise.


- Providing some experiments that test how the new approach is performing credit assignment. [Reviewer YcBP]

My evaluation: The authors added a new Section 5.3 to address this issue.



- Running the experiments on the latest version of Starcraft Multi-agent Challenge. [Reviewer uvcs]

My evaluation: This is not a serious issue for the claims of this work.



- The paper initially claimed that LAN reaches SOTA on all maps, which is not correct. [Reviewer HmEE]

My evaluation: This is revised in the paper.



- One of the reviewers mentioned in the private discussion that a paper (Hu et al., "Rethinking the Implementation Tricks and Monotonicity Constraint in Cooperative Multi-Agent Reinforcement Learning," 2021) shows that QMIX with Adam performs much better than its original version that apparently used RMSProp. For example, QMIX-Adam can solve the corridor domain in about 5M steps, while QMIX-RMSProp cannot.

My evaluation: This is not a major issue, but I encourage the authors to consider the refined version of QMIX as well.




In addition to these concerns by the reviewers, I have some other concerns:

- Related Work appears in Section 3, before the method is explained. It compares LAN with other methods. At this point in the paper, however, the reader does not understand LAN well enough to appreciate its differences with other methods, so comparison is not very meaningful. I realize this new position for the Related Work section was due to the request of Reviewer 3z8K. Their point was that comparing with other methods in the Experiments section would be confusing if the baselines are not described yet. That is a valid point. But just shifting the Related Work does not work either. This reduces the readability of the paper.


- The sentence "LAN can represent all decentralized policies" on page 4 is unclear and requires a clarification.


- The equation of P_a at the top of page 5 is inaccurate. The extended state $\tilde{s}'$ has both $s'$ and $\tau_{-a}'$ in it, but the RHS of that equation does not show what the probability of $\tau_{-a}'$ is.
Comparing with Equation (5) of Foerster et al. (2017), we see that some terms are missing.

- The Bellman equation of $Q^{\pi_a}(\tau_a, u_a)$ on the same page has $P(\tilde{s} | \tau_a)$ on the RHS. What is $P(\tilde{s} | \tau_a)$? The transition probability $P$ is defined over the state $s$, and not $\tau_a$.
I understand what the intention is, but the notation is not clear as is.
Foerster et al. (2017) uses $P$ and $p$ with different meanings.


- What is the target value for learning $V^\pi$? Does each agent learn $\tilde{Q}_a$ (using Equation (5) as the target) or $A_a$ (using Equation (3) as the target)?


- The discussion around bias and variance just after Equation (4) confusing. First, it is mentioned that "using a centralized target to learn a decentralized object might lead to high variance updates", but later "In LAN, the value is centralized while the Advantage is decentralized. ... This leads to a bias, but in turn also decreases the variance."

Does it increase variance or decrease it?

Moreover, can we have a closer inspection of these effects?
I understand that Figure 7 in Appendix E provides the gradient norm of updates, but perhaps one can study them more carefully and closely?

- The centralized value network uses the sum of embeddings. Although this ensures that the dimension of the input to the value network is independent of the number of agent, it leads to the information loss because of the use of summation. Can the authors expand on this and perhaps compare it with some alternatives, such as concatenating the embeddings?

- The paper mentions that LAN is more scalable than other methods in terms of the number of parameters. Isn't the main issue of scalability the sample complexity and not the number of parameters, especially as all the networks in this work are relatively small, ranging from 30K to 700K parameters, according to Tables 1 and 2.

- The authors use $1.25e^6$ and similar notations. I believe they mean $1.25 \times 10^6$, and not the Euler number to the power of 6. A more standard notation is $1.25E6$.

**Audience:**

Multi-agent RL is within the scope of TMLR. All reviewers agree with this.

**Claims And Evidence:**

The major claim that the method performs comparable to other state of the art methods and performs much better in some difficult domains have enough empirical evidence, though there is room for better statistical analysis.
The claims about how and why the method actually performs well are less substantiated and requires more evidence or change of the certainty of claims.